# Real-Life Cefiderocol Use in Bone and Joint Infection: A French National Cohort

**DOI:** 10.3390/antibiotics14040388

**Published:** 2025-04-08

**Authors:** Ava Diarra, Maxime Degrendel, Isabelle Eberl, Tristan Ferry, Karim Jaffal, Lelia Escaut, Antoine Asquier Khati, Nicolas Taar, Johan Courjon, Laurène Deconinck, Benjamin Lefevre, Aurélie Baldolli, Messaline Bermejo, Alexandre Bleibtreu, Vincent Dacquet, Victoire de Lastours, Pierre Gazeau, Romaric Larcher, Pierre Patoz, Olivier Robineau, Nicolas Rouzic, Naomi Sayre, Diana Isabela Costescu Strachinaru, Benjamin Valentin, Heidi Wille, Eric Senneville

**Affiliations:** 1Department of Infectious Diseases, Hôpital Gustave Dron, 59200 Tourcoing, France; 2Department of Infectious Diseases, Hôpital Avicenne, 75015 Paris, France; 3Department of Statistics, Hôpital Gustave Dron, 59200 Tourcoing, France; 4Infectious Diseases Department, Dijon Bourgogne University Hospital, 21079 Dijon, France; 5Service des Maladies Infectieuses et Tropicales, Hospices Civils de Lyon, Groupement Hospitalier Nord, Université Claude Bernard Lyon 1, 69004 Lyon, France; 6Centre International de Recherche en Infectiologie, CIRI, Inserm U1111, CNRS UMR5308, ENS de Lyon, UCBL1, 69007 Lyon, France; 7Infectious Diseases Department, Raymond-Poincaré University Hospital, 92380 Garches, France; 8Department of Infectious Diseases, Hôpital de Bicêtre, AP-HP, 94270 Le Kremlin-Bicêtre, France; 9Infectious Diseases Departement, Centre Hospitalier Universitaire de Nantes, 44000 Nantes, France; 10Infectious Diseases Departement, Valenciennes Hospital, 59300 Valenciennes, France; 11Infectiologie, Hôpital de l’Archet, Centre Hospitalier Universitaire de Nice, Université Côte d’Azur, 06204 Nice, France; 12U1065, Centre Méditerranéen de Médecine Moléculaire, C3M, Virulence Microbienne et Signalisation Inflammatoire, Inserm, 06204 Nice, France; 13Service des Maladies Infectieuses et Tropicales, Hôpital Bichat-Claude Bernard, AP-HP, 75018 Paris, France; 14Service des Maladies Infectieuses et Tropicales, Université de Lorraine, CHRU-Nancy, 54000 Nancy, France; 15Inserm, INSPIIRE, Université de Lorraine, 54000 Nancy, France; 16CHU de Caen, Infectious Diseases Department, 14000 Caen, France; 17Department of Internal Medicine and Infectious Diseases, Troyes Hospital, 10000 Troyes, France; 18Department of Infectious Diseases, Pitié-Salpêtrière Hospital, Assistance Publique-Hôpitaux de Paris (AP-HP), Sorbonne Université, 75018 Paris, France; 19Infectious Disease Unit, Centre Calot, Fondation Hopale, 62600 Berck-sur-Mer, France; 20Internal Medicine Department, Beaujon University Hospital, AP-HP, University of Paris, 92110 Clichy, France; 21Infectious Diseases and Tropical Medicine, La Cavale Blanche University Hospital, 29200 Brest, France; 22Department of Infectious and Tropical Diseases, PhyMedExp (Physiology and Experimental Medicine), Inserm (French Institute of Health and Medical Research), CNRS (French National Centre for Scientific Research), University of Montpellier, Nimes University Hospital, 30900 Nimes, France; 23Department of Bacteriology, Hôpital Gustave Dron, 59200 Tourcoing, France; 24Institut Pierre-Louis d’Epidémiologie et de Santé Publique, Inserm, Sorbonne Université, 75018 Paris, France; 25EA2694, Université Lille, Centre Hospitalier de Tourcoing, 59200 Tourcoing, France; 26Infectious Diseases Departement, Centre Hospitalier de Lorient, 56322 Lorient, France; 27Service de Maladies Infectieuses et Tropicales, Hôpital Delafontaine, 93200 Saint Denis, France; 28Department of Infectious Disease, Reine Astrid Hospital of Brussels, 1120 Bruxelles, Belgium; 29Department of Pharmacology, Lille University Hospital, 59000 Lille, France; 30Infectious Diseases Department, Centre Hospitalier de la Côte Basque, 64109 Bayonne, France

**Keywords:** cefiderocol, bone and joint infection, Gram-negative bacilli, tolerance, outcome

## Abstract

**Background**: Cefiderocol (CFD) is a novel siderophore cephalosporin developed for the treatment of infections involving multidrug-resistant (MDR) Gram-negative bacilli (GNB) infections (1–3). For bone and joint infections (BJIs), the use of CFD is currently neither part of its market authorization nor recommended, and has not yet been assessed by large-scale studies. **Objectives**: To fill the scarcity of data regarding the use of CFD in BJIs, we aimed to describe patients’ and infection characteristics along with the outcomes of the infection. **Methods**: We conducted a retrospective observational multicenter study in 22 French centers from January 2019 to December 2023. **Results**: From January 2019 to December 2023, 45 patients were included. Patients were mainly males (73%) with a median age of 62 years (interquartile range [IQR] 29), and a median Charlson comorbidity index of 3. Implant-related infections (20) were the most prominent, accounting for 44% of the cases. Carbapenemase-producing GNB were involved in 74% of the cases (*n* = 17/23), among which Pseudomonas aeruginosa accounted for 38% of these cases. Most patients received 6 g of CFD per day. CFD was used in combination with an antibiotic in 40 out of 45 cases (89%). The median duration of CFD treatment was 34 days. Seven patients (16%) experienced side effects, mainly gastro-intestinal disorders, including three (7%) who induced treatment cessation. Infection control included surgery in 37 (82%) patients. Failures and deaths occurred, respectively, in 22 (49%) and 10 (22%) cases. **Conclusions**: Our results suggest that CFD may be an alternative in MDR-GNB infections with limited therapeutic options.

## 1. Introduction

Cefiderocol (CFD) is a novel siderophore cephalosporin developed for the treatment of infection involving multidrug-resistant (MDR) Gram-negative bacilli (GNB) infections, in particular carbapenem-resistant strains [1]. The structural characteristics of CFD show similarity to both ceftazidime and cefepime, which enable CFD to overcome hydrolysis by β-lactamases, with the addition of a catechol moiety on the C-3 side chain, which chelates iron and mimics naturally occurring siderophore molecules [2]. CFD has demonstrated structural stability against hydrolysis by all four classes of beta-lactamases [3,4,5]. Its efficacy and safety on nosocomial pneumonia, bloodstream infections, and complicated urinary tract infections were previously demonstrated in randomized multicentric studies [6,7,8]. CFD use for bone and joint infections (BJIs) is currently neither part of its market authorization nor recommended, and is not validated through a randomized study. Approximately 17% of BJIs involve GNB, including Enterobacterales and non-fermenting GNB [9]. According to the Antimicrobial Resistance Surveillance Network (EARS-Net), 11% of *Escherichia coli* and *Klebsiella pneumoniae* and 19% of *Pseudomonas aeruginosa* were carbapenem-resistant in 2022 [10]. Data regarding the tolerability of the available antibiotic regimens for MDR-GNB infections as prolonged therapy are lacking.

One particular concern in the setting of BJIs is bacterial biofilm development [11]. In the biofilm setting, where antibiotic resistance is high but iron scavenging is important, CFD may have advantageous antimicrobial properties. An in vitro study has tested the efficacy of CFD versus other antibiotics on the reduction in biofilm and has shown that CFD effectively reduces biofilm and is a potent inhibitor of planktonic growth across a range of medically important GNB [12].

The few cases published about CFD for the treatment of BJIs are limited to rescue therapy or compassionate use [13,14,15,16,17,18,19]. Our study aimed to describe a large series of patients and infection characteristics along with outcomes. Our secondary objective was to fill the gap in CFD tolerance over long-term use.

## 2. Results

Demographics

From January 2019 to December 2023, 45 patients were included from 21 centers, mostly from France. Demographics and comorbidities are depicted in Table 1, noting that the median Charlson comorbidity index was 3 (interquartile range [IQR 5] [20]. Among patients with implant-related infections, 12 had not undergone any previous intervention. When disclosed, three of those patients originated from Mali, Tunisia, and the Republic of Madagascar.

A total of 115 microbiological samples were obtained, mostly intra-operatively, resulting in the identification of 29 different pathogens (details in Table 2. Four were of unknown origin (Figure 1). Among the 32 (71%) polymicrobial infections, Gram-positive bacteria were involved in 12 of these cases (37.5%) and anaerobes in 8 (25%). MDR GNB motivating CFD prescription are described Figure 2. The antibiotic susceptibility profile of bacterial isolates was available for 40 (89%) patients. Data about resistance mechanisms were available for 23 GNB (51%) and are shown in Figure 3. Carbapenemase-producing isolates were the most prevalent (see Figure 3), accounting for 17 (74%), including 5 New Delhi Metallo-beta-lactamase (NDM), 4 Verona Integron-encoded Metallo-beta-lactamase (VIM), and 4 oxacillinases (OXA). Susceptibility to the new beta-lactam and beta-lactamase inhibitor combinations was as follows: ceftazidime–avibactam (4/23, 17%), ceftolozane–tazobactam (3/23, 13%), meropenem–vaborbactam (5/13, 38%), and imipenem–relebactam (3/9, 33%).

CFD monitoring and use

Prior to CFD treatment, a median of two antibiotic lines were administered, including empirical antibiotic therapy in two cases, based on prior documentation. For culture-directed treatment, CFD was administered in combination in 40 cases out of 45 (89%) with other antibiotics to (i) cover micro-organisms out of the spectrum of CFD (21/45, 47%) or (ii) potentiate the antibacterial activity against GNB (31/45, 69%) (Table 3 and Figure 4). The median number of antibiotics associated with CFD was 2 (IQR 1).

CFD MICs ranged from 0.01 to 8 mg/L (median 0.5 mg/L (IQR 2). Details are shown in Appendix A. In one patient initially treated with CFD in combination with colistin, CFD had to be stopped when the French referral center reported resistance to CFD (*Acinetobacter baumannii*).

Three patients (7%) experienced side effects that resulted in CFD withdrawal, including stage 4 diarrhea, unknown stage acute kidney failure, and stage 2 hyper-eosinophilia.

Side effects

Seven (16%) CFD-induced side effects, according to the clinician’s opinion, included gastro-intestinal disorders (four, 57%), acute renal insufficiency (one, 14%), rhabdomyolysis (one, 14%), and skin reaction (one, 14%). In four cases out of seven, side effects occurred while CFD was associated with colistin and colistin plus tigecycline in another case. Most side effects were mild grade 2 (median grade 2, IQR 1). The most severe one (grade 4 diarrhea) was recorded in a patient treated with the CFD–daptomycin combination.

Infection management and outcomes

Surgery was performed in 19 of the 20 (95%) implant-related infections, including removal of the implant in 12 cases (63%) and in 18 of the 25 (72%) patients with osteomyelitis. The median delay between infection and surgery was 71 days (IQR 393). Details on surgery are available in Table 4.

Twenty-two (49%) patients experienced failure, which was microbiologically documented in fourteen (63%) cases. Pathogens similar to the initial one, according to gender, species, and antibiotic susceptibility patterns, were identified in seven (50%) of these cases. Four fungal superinfections occurred in the absence of prior antifungal treatment. Five superinfections involved *Staphylococcus* spp. despite targeted antibiotic treatment in four of these cases. Among the six failure patients for whom CFD susceptibility data were available, no resistance to CFD emerged during treatment for the five pathogens found identical in the initial infection and in revision surgery.

The median duration of follow-up was 291 days (IQR 339) with a minimum of 35 days. Ten (36%) patients had 2 years of follow-up available. During follow-up, 10 (22%) deaths were reported, including 7 (16%) related to vertebral osteomyelitis (*n* = 1), other osteomyelitis (*n* = 2), and implant-related infection (*n* = 4). Those deaths occurred with a median of 67 days following CFD initiation (IQR 247). Among 23 patients in remission, 13 had a known end of follow-up period with a median of 350 days (IQR 326).

No significant differences were shown between remission and failures, in terms of comorbidities, infection type, management in terms of antibiotic therapy and surgery, hence no predictive factor of failure was isolated (Table 5).

## 3. Discussion

We present herein, to our knowledge, the first study that focuses on CFD use for BJIs. Most of our patients were older, overweight males with multiple comorbidities, a predominance of orthopedic device-related infections, and most infections were polymicrobial (Table 1). The high rate of failure (49%) and deaths (22%) reported in our patients contrasts with the studies reported so far on CDF use for BJIs. Of note, these studies were all case reports. Table 6 sums up previous published cases of CFD use for BJIs. Among seven cases of osteomyelitis and prosthetic-related infections, CFD was combined in three cases, and all of the treatment regimens resulted in remission. In the present series of 45 patients, complex BJIs affecting frail and comorbid patients may explain our results.

BJIs involving extensively drug-resistant *P. aeruginosa* and carbapenemase (metallo-beta-lactamase)-producing isolates were the main reasons for choosing CFD. Of note, a small proportion of pathogens showed sensitivity to other antibiotics, which raises the question of why CFD was chosen over those alternatives, which was not clearly reported by the investigators of our study. The second most prevalent bacterium was *A. baumannii,* which has been associated in a large randomized trial on pulmonary and bloodstream infections with higher mortality in patients treated with CFD compared to the standard of care [6]. These data were, however, not confirmed in another trial on pneumonia [7].

In our series of patients, most CFD prescriptions comprised combinations with another anti-MDR-GNB antibiotic with the aim of optimizing the antibacterial activity. Indeed, these difficult-to-treat infections are characterized by limited antibiotic site penetration and bacterial persistence in the biofilm environment. Combination antibiotic regimens may also help prevent the occurrence of resistance, although only limited data support this statement. Current guidelines recommend the use of CFD within a combination, at least for carbapenem-resistant *A. baumannii* infections [21]. Of note, a recent real-life Italian study found no differences in terms of mortality between CFD monotherapy and combination in infections other than BJIs [22].

Seven (16%) patients experienced side effects, mainly with minor impact. The most prominent was gastro-intestinal disorder, which did not result in the withdrawal of the treatment. However, the nearly systematic combination with other antibiotics made it difficult to attribute the adverse effects to CFD. Nonetheless, CFD was administered for a median period of 4 weeks, advocating for good tolerability among patients even in the case of prolonged treatment. Of note, one patient was given CFD for a total duration of 4 months without reporting any adverse event attributable to CFD. None of our patients experienced anemia, as it has already been reported in another study in relation to iron deficiency in a patient treated with prolonged CFD therapy [18].

Ten patients (22%) died by the end of the follow-up, and the majority of those deaths were related to the infection, which may reflect the patient’s frailty, but also the high failure rates usually reported in patients with MDR GNB BJIs [23]. The part of a CFD’s insufficient efficacy in these bad outcomes could not be evaluated in our study. Nearly half of those failures were related to the same micro-organism, which had motivated the use of CFD, noting that the acquisition of resistance to CFD during treatment was recorded in only one of these cases.

We could collect the data on therapeutic drug monitoring (TDM) for CFD in six (13%) patients for whom all trough plasma concentrations were at least four times higher than the MIC of the pathogen that had motivated the use of CFD. TDM could provide a better understanding of CFD efficiency and toxicity. Only a few studies reported the results of CFD TDM (trough plasma concentrations) [24,25]. Prinz et al. measured a median (interquartile range) value of 50.0 (27.2–74.6) mg/L in five critically ill patients. Gatti et al. reported values for the free fraction ranging from 0.59 to 56.78 mg/L in 13 patients, while Schellong et al. reported a value of 6.84 mg/L in a patient treated for osteomyelitis. Therapeutic targets in terms of CFD plasma concentrations and data about bone penetration and biofilm deserve additional investigations.

Our study holds several limitations. Firstly, the retrospective collection of data induces multiple unavoidable biases. Of note, we collected no additional data on CFD administration (continuous or intermittent infusion), and we could not evaluate practices or potential administration in an outpatient setting. Data generalizability concerning MIC is impaired by the availability of a micro-dilution kit, by the infection time, and the variability of techniques according to the pathogens isolated per infection. A prospective cohort design was not intended due to the scarcity of CFD prescriptions for BJIs per center per year. Secondly, BJIs recover a wide range of infections, from vertebral osteomyelitis to orthopedic device-linked infection, which may impact the reliability of our results. Finally, long-term follow-up data were lacking.

## 4. Materials and Methods

This retrospective observational study was conducted in 21 French and 1 Belgian center from January 2019 to December 2023. French centers were targeted thanks to the French infectious disease network. The patients included were adults with GNB-related BJIs treated with CFD regardless of the duration of treatment, provided they had received at least 4 days of treatment. The exclusion criteria were patients under 18 years of age, adults under legal guardianship, and those who expressed written opposition. Patients’ data were collected until the date of the latest news available. Our first objective was to describe the population’s characteristics and comorbidities and to depict the pathogens involved in BJIs treated with CFD. The secondary objective was to evaluate the clinical outcome of treatment with CFD in patients with GNB-related BJIs.

### 4.1. Definitions

Remission was defined as the absence of signs of infection at the initial site at the end of follow-up. Failure was defined as recurrence, relapse, superinfection, and any other situation other than remission, including BJI-related death. Recurrence and relapse were defined, respectively, as the occurrence of infection at the same site involving the same bacteria within 6 months and more than 6 months after the initial joint infection. Superinfection was defined as the occurrence of an infection at the same location due to pathogen(s) distinct from the initial one [26].

Adverse events were described and analyzed according to the grades as follows: grade 1 (mild), grade 2 (moderate), grade 3 (severe), grade 4 (life-threatening or disabling) events, and grade 5 (resulting in death).

Minimum inhibitory concentrations (MICs) were determined using micro-dilution (UMIC^®^, COMASP^®^) and diffusion according to the center appreciation [27,28].

### 4.2. Statistical Analysis

Categorical variables are expressed in terms of frequency and percentage. Quantitative variables are represented as means ± standard deviation (SD) or medians (med) and interquartile range (IQR), depending on their normality. Comparisons were performed using the Chi-square test or Fisher’s exact test for categorical variables and the Wilcoxon test for quantitative variables. Statistical testing was conducted at the two-tailed α-level of 0.05. Data were analyzed using the R (R Statistical Software (v4.4.1; R Core Team 2024)) [29].

## 5. Conclusions

As suggested in the present retrospective study, CFD may be useful as salvage therapy for patients with BJIs and limited treatment options due to antimicrobial resistance and/or drug-related toxicity. The good profile of tolerance of CFD therapy, even in prolonged administration, warrants additional data. The high failure rate reported in our patients was not associated with the emergence of resistance to CFD and is likely to be in relation to patients’ co-morbidities and frailty.

## Figures and Tables

**Figure 1 antibiotics-14-00388-f001:**
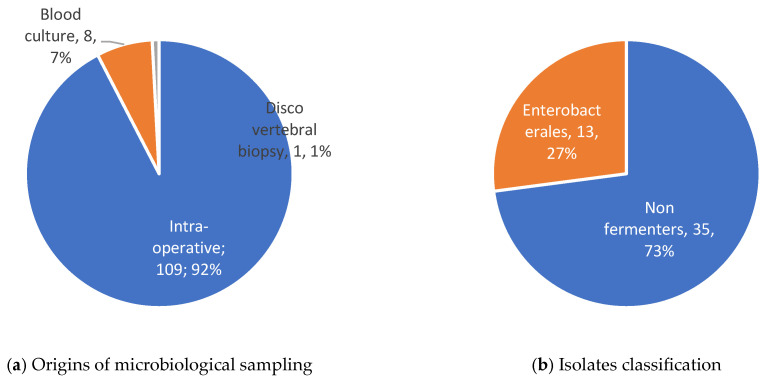
Origins of microbiological sampling (**a**) and isolates classification (**b**).

**Figure 2 antibiotics-14-00388-f002:**
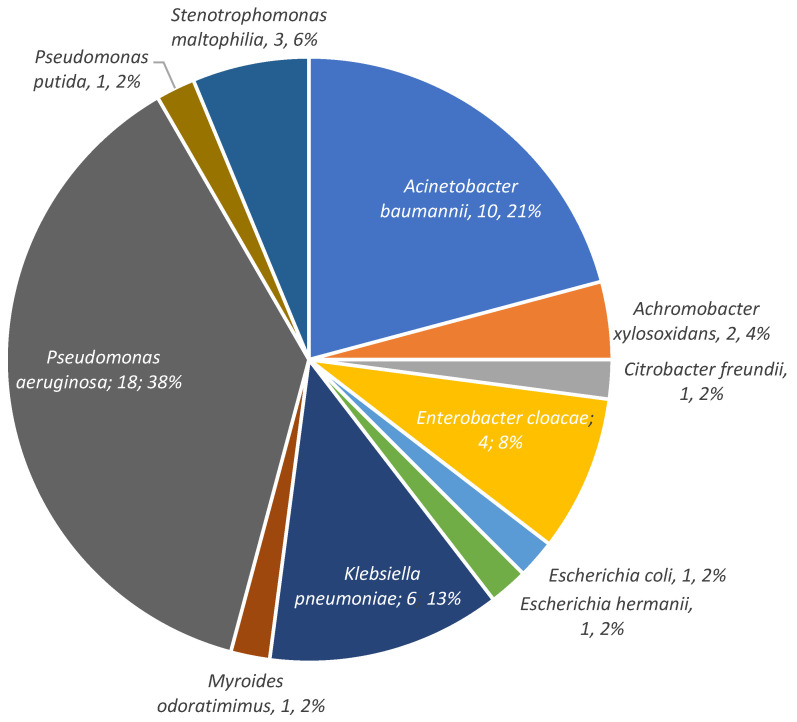
Isolates in detail.

**Figure 3 antibiotics-14-00388-f003:**
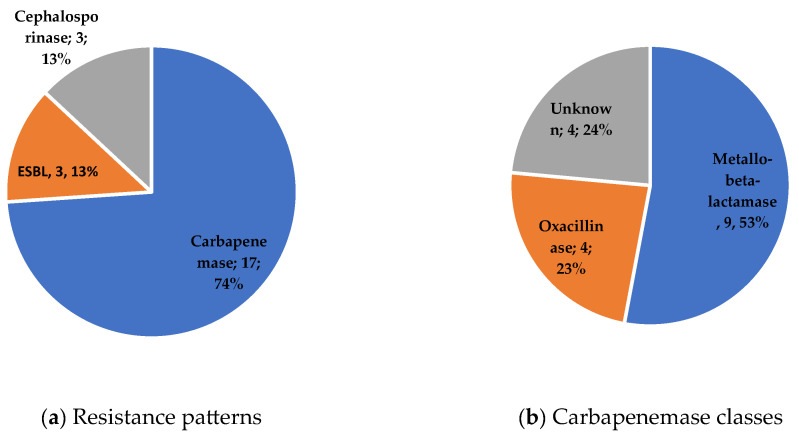
Resistance patterns (**a**) and carbapenemase classes (**b**) for 23 and 17 GNB strains, respectively.

**Figure 4 antibiotics-14-00388-f004:**
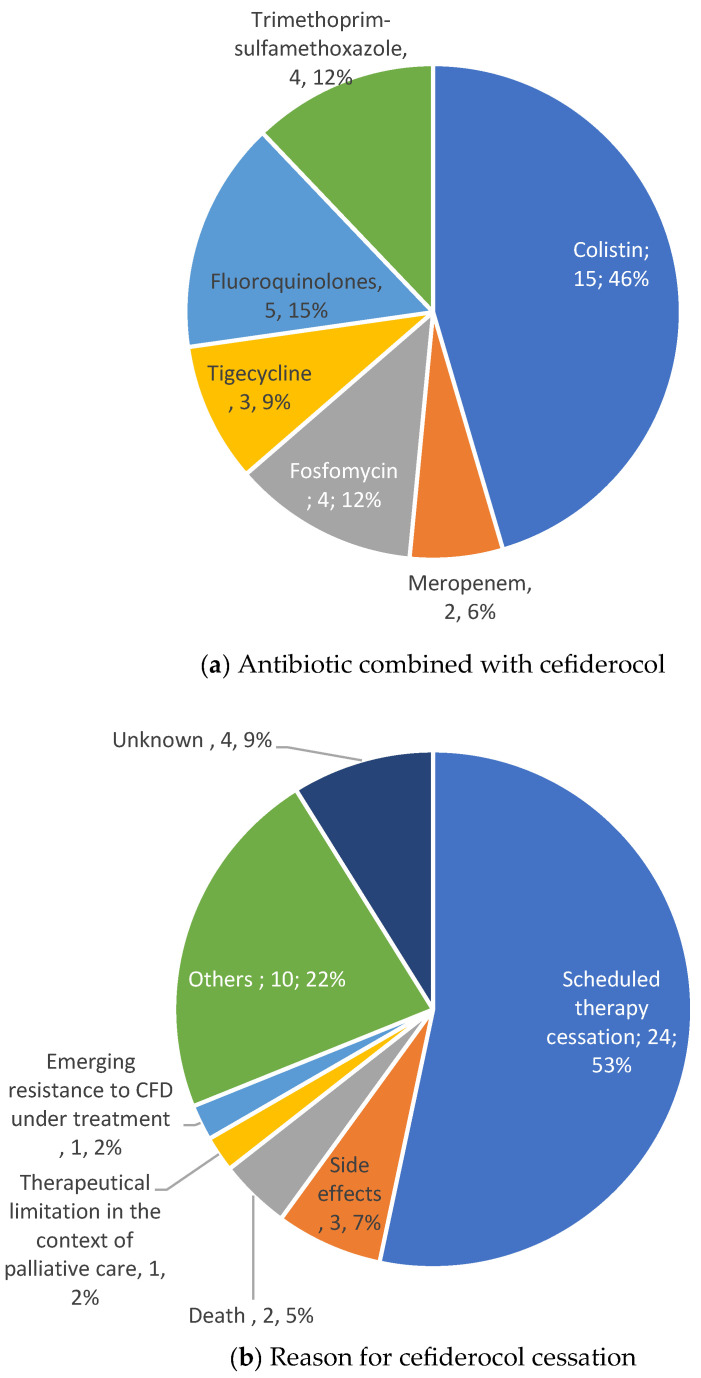
Antibiotic combined with cefiderocol (**a**) and reason for cefiderocol cessation (**b**).

**Table 1 antibiotics-14-00388-t001:** Patient’s demographics, comorbidities, and infection presentation.

Demographics, Comorbidities, and Infection Presentation		*n* = 45 [Number of Unknown Cases]
Patient’s characteristics	Age, years (median ± IQR)	62 ± 29 [1]
	Sex *n* (%)	
	Male	33 (73%)
	Female	12 (27%)
	BMI, kg/m^2^ (median ± IQR)	25 ± 9 [3]
	Charlson comorbidity index (median ± IQR)	3 ± 5 [1]
	ASA score (median ± IQR)	3 ± 1 [20]
Infection	Osteomyelitis *n* (%)	25 (56%)
	Vertebral osteomyelitis	4 (9%)
	Diabetes-related osteomyelitis of the foot	5 (11%)
	Other	16 (36%)
	Implant-related infections *n* (%)	20 (44%)
	Prosthetic joint *	8 (18%)
	Arthrodesis	3 (7%)
	Osteosynthesis	9 (20%)
	Number of previous surgical intervention(s) in patients with implant-related infections (median ± IQR)	0 ± 2
	Duration between surgery and implant-related infections out of 20 (days) [median ± IQR]	71 ± 393
	<1 month	9 (43%)
	≥1 month	12 (57%)
	<3 months	17 (57%)
	[3–24 months]	9 (30%)
	≥24 months	4 (13%)
Clinical manifestations	Temperature > 38.5°	17 (40%) [2]
	Fistula	19 (44%) [2]
	Pain	30 (70%) [2]
	Local signs of inflammation	34 (76%)
	Duration between symptoms of infection and diagnosis of infection (days) (median ± IQR)	20 ± 37
Biological markers (median ± IQR)	Leucocytes (G/L)	8.08 ± 6.62 [1]
	Polymorph neutrophils (G/L)	3.34 ± 5.46 [3]
	CRP (mg/L)	22 ± 128 [4]
	Blood Creatinine (µmol/L)	78 ± 59 [2]
	GFR (mL/min)	83 ± 41 [1]

Abbreviations: ASA: American Society of Anaesthesiologists (19); BMI: body mass index; CRP: C-reactive protein; GFR: glomerular filtration rate. * 7 knee and 1 hip prosthetic device

**Table 2 antibiotics-14-00388-t002:** An exhaustive list of isolates.

*Achromobacter xylosoxidans*	3 (2.6%)
*Acinetobacter baumanii*	10 (8.7%)
*Bacteroides ovatus*/*xylanisolvens*	1 (0.9%)
*Candida albicans*	1 (0.9%)
*Candida orthopsilosis*	1 (0.9%)
*Citrobacter freundii*	4 (3.5%)
*Clostridium sporgenes*	1 (0.9%)
*Corynebacterium striatum*	1 (0.9%)
*Cutibacterium acnes*	5 (4.3%)
*Enterobacter* spp.	6 (5.2%)
*Enterococcus* spp.	12 (10%)
*Escherichia coli*	6 (5.2%)
*Escherichia hermannii*	1 (0.9%)
*Globicatella sanguinis*	1 (0.9%)
*Klebsiella pneumoniae*	10 (8.7%)
*Lactobacillus* spp.	1 (0.9%)
*Morganella morganii*	2 (1.7%)
*Myroides odoratimimus*	1 (0.9%)
*Proteus mirabilis*	4 (3.5%)
*Pseudomonas aeruginosa*	18 (24%)
*Pseudomonas putida*	1 (0.9%)
*Staphylococcus aureus*	2 (1.7%)
*Staphylococcus epidermidis*	5 (4.3%)
*staphylococcus haemolyticus*	2 (1.7%)
*Staphylococcus warneri*	1 (0.9%)
*Stenotrophomonas maltophilia*	4 (3.5%)
*Terrisporobacter glycolius*	1 (0.9%)

**Table 3 antibiotics-14-00388-t003:** Cefiderocol monitoring and use.

Cefiderocol Monitoring and Use		*n* = 45
Cefiderocol dosage, *n* (%)	0.75 g bid *	1 (2%)
	1 g tid *	1 (2%)
	1.5 g tid	2 (4%)
	2 g tid	40 (89%)
	2 g qid *	1 (2%)
Cefiderocol monitoring	Cefiderocol blood monitoring, *n* (%)	6 (14%)
	Cefiderocol plasma trough concentration, mg/L (median ± IQR)	17.4 ± 10
Duration of treatment	Duration of cefiderocol treatment, days (median ± IQR) [unknown]	34 ± 47 [6]
	Range	4–122

* bid: “bis in die” two times a day, tid: “ter in die” three times a day, qid: “quarter in die” four times a day.

**Table 4 antibiotics-14-00388-t004:** Infection management and performed surgery.

Management and Surgery	*n* = 45
**Surgery [*n* (%)]**	37 (82%)
Orthopedic device (% out of infection devices linked)	19 (95%)
On infection without an orthopedic device (% out of infection without a device)	18 (72%)
**Surgery on infection involving orthopedic device [*n* (%)]**	*n* = 19
**Device’s withdrawal**	12 (63%)
One-stage exchange	6 (32%)
Two-stage exchange	2 (11%)
Arthrodesis	1 (5%)
Removal without replacement	1 (5%)
Other	2 (11%)
**Device’s retention/DAIR**	7 (37%)
Other	1 (5%)
**Surgery on infection without orthopedic device [*n* (%)]**	*n* = 18
Irrigation lavage	9 (50%)
Bone resection	4 (22%)
Amputation	3 (17%)
Other	2 (11%)

Abbreviations: DAIR: debridement antibiotics and implant retention.

**Table 5 antibiotics-14-00388-t005:** Variables associated with patients’ outcomes.

Comparison Between Success and Failure		Remission (*n* = 23, 51%)	Failure (*n* = 22, 49%)	*p*-Value
Comorbidities	Age, years (median, Q1, Q3)	62 (49–68)	62 (36–74)	0.8
	Gender male/female, *n* (%)	18 (78%)/5 (22%)	15 (68%)/7 (32%)	0.4
	BMI, kg/m^2^ (median, Q1, Q3)	26 (24–32)	24 (20–11)	0.2
	Charlson comorbidity index (median, Q1, Q3)	3 (1–7)	4 (2–6)	0.7
Type of infection	Infection involving orthopedic device *n* (%)	9 (39%)	11 (50%)	0.6
Clinical presentation	Local signs of inflammation, *n* (%)	15 (65%)	19 (86%)	0.1
	Duration between symptoms and diagnosis, days (median, Q1, Q3)	26 (6–61)	12 (3–27)	0.2
Microbiology	Enterobacterales *n* (%)	12 (18%)	5 (10%)	0.7
	Non-fermenters *n* (%)	25 (38%)	22 (44%)	0.7
	Fungi *n* (%)	2 (3%)	0	0.7
	Carbapenemase-producing GNB *n*/23 (%)	10/13 (77%)	7/10 (70%)	0.7
	Polymicrobial infection *n* (%)	17 (74%)	15 (68%)	0.7
Antibiotic regimens	Combined anti-GNB therapy *n* (%)	15 (22%)	11 (19%)	0.8
	Duration of cefiderocol treatment, days (median, Q1, Q3)	22 (9–49)	42 (14–65)	0.2
Surgery	Removal of the implant device (*n*/19, %)	5 (56%)	7 (70%)	0.6
	Bone resection (*n*/18, %)	3 (38%)	1 (10%)	0.3
	Delay between infection onset and surgery days (median, Q1, Q3)	78 (21–411)	29 (9–631)	0.5

**Table 6 antibiotics-14-00388-t006:** Variables associated with patients’ outcomes.

Reference	Isolate	Type of Infection	Associated Antibiotic	Removal of the Infected Implants	Duration (Days)	Outcome (Adverse Effect)
Mabayoje et al., 2021 [13]	*Acinetobacter baumanii* NDM	Osteosynthesis	Tigecycline	Yes	25	Remission
Siméon et al., 2020 [14]	*Enterobacter hormaechei* Derepressed cephalosporinase and β-lactamases (CTX-M-15, TEM-1B and OXA-1)	Total knee prosthesis	___	No	84	Remission
Dagher et al., 2021 [15]	*Acinetobacter baumannii*	Osteomyelitis	___	No	109	Remission
Alamarat et al., 2020 [17]	*Pseudomonas aeruginosa bla*_NDM-1_ and ESBL-producing *Klebsiella pneumoniae*	Osteomyelitis	___	Yes	98	Remission
Schellong et al., 2023 [18]	*Pseudomonas aeruginosa*, CFD MIC = 0.38 mg/L trough CFD [ ] = 6.84 mg/L	Osteomyelitis	___	Yes	169 (including 63 as OPAT)	Remission (iron deficiency anemia)
Chambers et al., 2023 [19]	*Stenotrophomonas maltophilia*	Total knee prosthesis	TMP-SMX	Yes	56	Remission (20 months)
Chavda et al., 2021 [16]	*Pseudomonas aeruginosa*	Osteosynthesis	Ciprofloxacin (for *Morganella morganii*)	Yes	24	Remission

## Data Availability

The raw data supporting the conclusions of this article will be made available by the authors on request.

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
