# Peer review of "Real-Life Cefiderocol Use in Bone and Joint Infection: A French National Cohort"

_antibiotics, 2025, doi:10.3390/antibiotics14040388_

Round 1

Reviewer 1 Report

Comments and Suggestions for Authors

Introduction

How does this study fill the gap in understanding cefiderocol efficacy compared to combination therapies based on current state of the art?

Why focus on GNB infections when BJIs often involve polymicrobial environments? Answer in this section

Methods

a prospective cohort design would have been more beneficial for such an analysis. Acknowledge this limitation in the Methods section

Suggest a standardized protocol/guide for diagnosing and managing BJIs across different centers centers

Include justification for focusing on French centers exclusively.

Results

underline the importance of achieving therapeutic drug monitoring targets for cefiderocol

A discussion on the significance of high polymicrobial infection rates for treatment planning must be added

What is the the role of combination therapies in improving outcomes for MDR infections?

Create a relation between failure rates to specific comorbidities or infection types for deeper understanding

Author Response

Thank you very much for your thorough reviewing. We will try to answer point by point accordingly.

Introduction

How does this study fill the gap in understanding cefiderocol efficacy compared to combination therapies based on current state of the art?

It is the first study, to our knowledge, to focus on the use of cefiderocol in bone and joint infections. Current recommendations do not provide the best practice guidelines for bone and joint infections involving difficult to treat GNB where scare antibioctics options are available.

Why focus on GNB infections when BJIs often involve polymicrobial environments? Answer in this section

Our study first objective was to describe cefiderocol use, hence we focused on GNB. Cefiderocol does exhibit some activity against Gram-positive bacteria, although its primary target is Gram-negative pathogens. In polymicrobial infections, identifying the exact cause of treatment failure becomes more complex. Furthermore, cefiderocol, like some other antibiotics, can act against multiple pathogens and is often used in combination with other agents. This makes it challenging to determine which specific drug contributed to the observed therapeutic effect.

Methods

a prospective cohort design would have been more beneficial for such an analysis. Acknowledge this limitation in the Methods section

We added line 225 “A prospective cohort design was not intended due to the scarcity of CFD prescription.”

Suggest a standardized protocol/guide for diagnosing and managing BJIs across different centers centers

Infections were diagnosed and management were defined by each participating center. However, common trends were isolated as infections were defined by fever or inflammatory symptoms by the bone and joint sites and management complies with current guidelines. 

Include justification for focusing on French centers exclusively.

We added line 331, French centers were targeted thanks to the French infectious disease network.

Results

underline the importance of achieving therapeutic drug monitoring targets for cefiderocol

We added line 209 “TDM could provide a better understanding of CFD efficiency and toxicity”.

A discussion on the significance of high polymicrobial infection rates for treatment planning must be added

Combination therapy was primarily driven by the multiplicity of pathogens involved in those complex infections, as shown with the rate of gram positive cocci, fungi and anaerobes. CFD, due to its wide spectrum may be of great interest in thiese situations.

What is the the role of combination therapies in improving outcomes for MDR infections?

The role of therapeutic combinations is to limit the emergence of resistance, enhance synergistic effects, and improve efficacy in environments where bacteria have altered metabolisms. This has been extensively studied in prosthetic joint infections (PJI) for Gram-positive bacteria but much less so for Gram-negative bacteria, except for Pseudomonas aeruginosa. Rationalizing combinations in polymicrobial infections involving Gram-negative bacteria remains a significant challenge.

Create a relation between failure rates to specific comorbidities or infection types for deeper

understanding

Unfortunately, no link could be driven between comorbidities and outcomes nor infection types. However we could hypothesize that the poor outcomes we identify in this study might be partially linked to the high level of comorbidities among the patients cohorts.

Reviewer 2 Report

Comments and Suggestions for Authors

The numbers of sample is too small, and the results obtained may be inaccurate. It is recommended to increase the sample size and then do analysis.

1. Line 55, “6g”shuold be “6 g.

2. Line 53-55,Carbapenemase-expressing GNB were involved in (n=17/23,74% of the cases) including Pseudomonas aeruginosa in 38%. This sentence is not smooth, as if something is missing.

3. Data from the Supplementary Material can be placed directly into the main text.

4. Adjust the text in Figure 1 to the same font size. Figure 1 title is placed below Fig. (A) in the Figure 1 header is not indicated in Fig. The sum of the numbers in (b) does not equal 23.

5. Table 1-3 are not beautiful and should be rearranged.

6. The result part should be divided into several meanings.

7. "table 5" is not marked in the article.

8. In table 4, The p-values are all greater than 0.05. In my opinion, the results of this analysis are meaningless. There is no need to write about this part.

9. The goal in the abstract is not equal to the paper title, and the content contained in the title is not presented in the results. Suggestion: Based on the existing results, re-refine the topic and research objectives.

Author Response

Thank you very much for your thorough reviewing. We will try to answer point by point accordingly.

  1. Line 55, “6g”shuold be “6 g”.

We corrected accordingly line 61

  1. Line 53-55,Carbapenemase-expressing GNB were involved in (n=17/23,74% of the cases) including Pseudomonas aeruginosa in 38%. This sentence is not smooth, as if something is missing.

We corrected line 59 « Carbapenemase-producing  GNB were involved in 74% of the cases (n=17/23,) among which Pseudomonas aeruginosa accounted for 38% of these cases.

  1. Data from the Supplementary Material can be placed directly into the main text.

We corrected accordingly

  1. Adjust the text in Figure 1 to the same font size. Figure 1 title is placed below Fig. (A) in the Figure 1 header is not indicated in Fig. The sum of the numbers in (b) does not equal 23.

We corrected accordingly. The Figure B focused on carbapenemase (n=17).

  1. Table 1-3 are not beautiful and should be rearranged.

My deepest apologies for that. We did not know how to rearrange precisely the tables. We are open to modify the table according to your liking.

  1. The result part should be divided into several meanings.

We divided the results into :

- Demographics

- CFD monitoring and use

- Side effects

- Infection management and outcomes

  1. "table 5" is not marked in the article.

We added line 172 Table 5 sums up previous published cases of CFD use for BJIs.

  1. In table 4, The p-values are all greater than 0.05. In my opinion, the results of this analysis are meaningless. There is no need to write about this part.

We agree that there is no difference between the groups in this table. However, this table outlines the characteristics of the patients, and it seems necessary for the reader to have this data for extrinsic comparison purposes in the study.

  1. The goal in the abstract is not equal to the paper title, and the content contained in the title is not presented in the results. Suggestion: Based on the existing results, re-refine the topic and research objectives.

We agree and change the name of the paper accordingly:

Cefiderocol real-life use in bone and joint infection: a French national cohort

Reviewer 3 Report

Comments and Suggestions for Authors

The authors conducted a retrospective study for Cefiderocol (CFD) across 22 French centers from 2019 to 2023, involving 45 patients, which highlighted the prevalence of implant-related infections and carbapenemase-expressing GNB. Most patients received CFD as part of combination therapy, with a median treatment duration of 34 days. Adverse effects were observed in 16% of cases, leading to treatment discontinuation in 7%. The study outcomes included a 49% failure rate and 22% mortality. The authors suggested that CFD could be a viable alternative for MDR-GNB infections with limited therapeutic options. While the manuscript is well-written and presents its findings effectively, several areas require improvement to enhance the overall quality of the research.

1.     Repetition in Lines 46–74 and 73–74

The text in these lines is identical. One instance should be rephrased to avoid redundancy.

2.     Rationale for Cefiderocol Selection

The introduction should include a clear rationale for choosing Cefiderocol over other antibiotics with similar antimicrobial profiles.

3.     Abbreviation Expansion

All abbreviations should be expanded upon their first occurrence in the manuscript. For example, the abbreviation IQR (Interquartile Range) in lines 52 and 92 should be defined.

4.     Accuracy of Figure 1 Description

Figure 1 depicts resistance patterns rather than mechanisms of resistance. The caption should be revised to accurately describe the figure.

5.     Inconsistency in Microbiological Samples

The manuscript states that 115 microbiological samples were collected (line 100), but Table 2 refers to only 111 samples. This discrepancy should be clarified, and Table 2 should be cited appropriately in the manuscript.

6.     Dosage Clarification in Table 3

Table 3 includes entries like "0.75 g tid," but it is unclear whether this refers to the dose per administration or the total daily dose. The authors should provide clarification.

7.     Attribution of Adverse Effects

The manuscript should explain how adverse effects were attributed to Cefiderocol rather than to other drugs when they were used in combination.

8.     Comparison with Existing Literature

The discussion should include a critical comparison of the study’s findings with those of existing literature to better contextualize the results and support the conclusions drawn.

Author Response

Thank you very much for your thorough reviewing. We will try to answer point by point accordingly.

The authors conducted a retrospective study for Cefiderocol (CFD) across 22 French centers from 2019 to 2023, involving 45 patients, which highlighted the prevalence of implant-related infections and carbapenemase-expressing GNB. Most patients received CFD as part of combination therapy, with a median treatment duration of 34 days. Adverse effects were observed in 16% of cases, leading to treatment discontinuation in 7%. The study outcomes included a 49% failure rate and 22% mortality. The authors suggested that CFD could be a viable alternative for MDR-GNB infections with limited therapeutic options. While the manuscript is well-written and presents its findings effectively, several areas require improvement to enhance the overall quality of the research.

  1. Repetition in Lines 46–74 and 73–74

The text in these lines is identical. One instance should be rephrased to avoid redundancy.

Please find line 81 « CFD use for bone and joint infections (BJIs), is currently neither part of its mar-ket authorization nor recommended and is not validated through randomized study

  1. Rationale for Cefiderocol Selection

The introduction should include a clear rationale for choosing Cefiderocol over other antibiotics with similar antimicrobial profiles.

We added line  96 “Our secondary objective was to fill the gap in CFD tolerance over long-term use”.

  1. Abbreviation Expansion

All abbreviations should be expanded upon their first occurrence in the manuscript. For example, the abbreviation IQR (Interquartile Range) in lines 52 and 92 should be defined.

We corrected accordingly

  1. Accuracy of Figure 1 Description

Figure 1 depicts resistance patterns rather than mechanisms of resistance. The caption should be revised to accurately describe the figure.

We corrected accordingly

  1. Inconsistency in Microbiological Samples

The manuscript states that 115 microbiological samples were collected (line 100), but Table 2 refers to only 111 samples. This discrepancy should be clarified, and Table 2 should be cited appropriately in the manuscript.

We added line 109 « Four were of unknown origin. »

  1. Dosage Clarification in Table 3

Table 3 includes entries like "0.75 g tid," but it is unclear whether this refers to the dose per administration or the total daily dose. The authors should provide clarification.

 We added line 139 « *bid: “bis in die” two times a day tid: “ter in die” three times a day qid: “quarter in die” four times a day »

  1. Attribution of Adverse Effects

The manuscript should explain how adverse effects were attributed to Cefiderocol rather than to other drugs when they were used in combination.

We considered the clinician point of view when attributing side effects. We only focused on side effects which the clinician linked directly to cefiderocol, hence inducing a bias.

  1. Comparison with Existing Literature

The discussion should include a critical comparison of the study’s findings with those of existing literature to better contextualize the results and support the conclusions drawn.

We added line 207 « Among 7 cases of osteomyelitis and prosthetic-related infections, CFD was combined in 3 cases and all of treatment regimen resulted in remission”

Round 2

Reviewer 2 Report

Comments and Suggestions for Authors

The explanation of the results is not clear enough, especially the format of the table is very messy. The sample size is not sufficient, the results may not be reliable, and the research significance is not significant. I think the research should still expand the sample size, such as collecting relevant data from other countries and conducting analysis. There are few figures, and the contents of Tables 2 and 3 can be represented by figures.

Author Response

The sample size is not sufficient, the results may not be reliable, and the research significance is not significant. I think the research should still expand the sample size, such as collecting relevant data from other countries and conducting analysis.

The study was designed for a specific time period and a defined number of centers. Due to time constraints and regulatory authorizations, it seems complex to extend the study to the European level following the same design. The objective of this study is to demonstrate the current use of Cefiderocol in France in the context of BJIs.

Published studies to date on cefiderocol outside its primary indications remain rare, and none focus on an organ-specific issue. We acknowledge that there is still some data heterogeneity when considering BJIs with and without implanted material. However, we emphasize that no other study contains as little heterogeneity as this one.

That being said, we agree that the next study should focus even more precisely on specific types of infections.

The explanation of the results is not clear enough, especially the format of the table is very messy.

There are few figures, and the contents of Tables 2 and 3 can be represented by figures.

We have rearranged the tables and converted some data into figures as suggested.
